# Overcoming Barriers in Hospital-Based Health Technology Assessment (HB-HTA): International Expert Panel Consensus

**DOI:** 10.3390/healthcare12090889

**Published:** 2024-04-25

**Authors:** Iga Lipska, Rossella Di Bidino, Maciej Niewada, Bertalan Nemeth, Tomasz Bochenek, Monika Kukla, Barbara Więckowska, Alicja Sobczak, Katarzyna Iłowiecka, Antal Zemplenyi, Nicolas Martelli, Tess Martin, Olena Filiniuk, Kostyantyn Kosyachenko, Rabia Sucu, Oresta Piniazhko, Olha Zaliska, Andrey Avdeyev, Nasrulla Shanazarov, Marina von Pinoci, Rok Hren

**Affiliations:** 1Health Policy Institute, 00-819 Warsaw, Poland; iga.lipska@healthpolicy.institute (I.L.); alicjateresa@gmail.com (A.S.); katarzyna.ilowiecka@healthpolicy.institute (K.I.); 2Medical Department, Academy of Applied Medical and Social Sciences, 82-300 Elbląg, Poland; 3Hospital-Based Health Technology Assessment Interest Group, Health Technology Assessment International, Edmonton, AB T6H 5P9, Canada; to.hospital-basedhta@htai.org (M.K.); rsucu@htai.org (R.S.); 4Fondazione Policlinico Universitario Agostino Gemelli IRCCS, 00168 Rome, Italy; rossella.dibidino@policlinicogemelli.it; 5Health Graduate School of Health Economics and Management (ALTEMS), Università Cattolica Del Sacro Cuore, 00168 Rome, Italy; 6Department of Experimental and Clinical Pharmacology, Medical University of Warsaw, 02-091 Warsaw, Poland; maciej.niewada@gmail.com; 7Syreon Research Institute, 1142 Budapest, Hungary; bertalan.nemeth@syreon.eu (B.N.); antal.zemplenyi@syreon.eu (A.Z.); 8Department of Nutrition and Drug Research, Institute of Public Health, Faculty of Health Sciences, Jagiellonian University Medical College, 31-126 Krakow, Poland; t.bochenek@uj.edu.pl; 9Social Insurance Department, Warsaw School of Economics, 02-554 Warsaw, Poland; barbara.wieckowska@sgh.waw.pl; 10Center for Health Technology Assessment and Pharmacoeconomics Research, Faculty of Pharmacy, University of Pécs, 7624 Pécs, Hungary; 11Pharmacy Department, Georges Pompidou European Hospital, 75015 Paris, France; nicolas.martelli@aphp.fr (N.M.); tess.martin@aphp.fr (T.M.); 12Faculty of Pharmacy, Université Paris-Saclay, GRADES, 91190 Orsay, France; 13Safe, Affordable, and Effective Medicines for Ukrainians (SAFEMed) Activity, Management Sciences for Health, 02000 Kyiv, Ukraine; olenafilinyuk@gmail.com; 14Department of Management and Economy of Pharmacy, Bogomolets National Medical University, 01601 Kyiv, Ukraine; kosleokos@gmail.com; 15Department of Health Technology Assessment, State Expert Center, Ministry of Health of Ukraine, 01021 Kyiv, Ukraine; orestapb@gmail.com; 16Department of Management and Economy of Pharmacy, Medicine Technology and Pharmacoeconomics, Faculty of Postgraduate Education, Danylo Halytsky Lviv National Medical University, 79010 Lviv, Ukraine; olzaliska@ukr.net; 17Medical Center Hospital of the President’s Affairs Administration of the Republic of Kazakhstan, Astana 010000, Kazakhstan; avdeyev.andrey@yahoo.com (A.A.); nasrulla@inbox.ru (N.S.); 18Care Quality Division, Hôpitaux Universitaires de Genève, 1211 Geneva, Switzerland; marina.von-pinoci@hug.ch; 19Institute of Mathematics, Physics, and Mechanics, 1000 Ljubljana, Slovenia; 20Faculty of Mathematics and Physics, University of Ljubljana, 1000 Ljubljana, Slovenia

**Keywords:** hospital-based health technology assessment, decision making in hospitals, facilitators influencing implementation, barriers influencing implementation

## Abstract

The purpose of this article is to investigate the common facilitators and barriers associated with the implementation of hospital-based health technology assessment (HB-HTA) across diverse hospital settings in seven countries. Through a two-round Delphi study, insights were gathered from a panel of 15 HTA specialists from France, Hungary, Italy, Kazakhstan, Poland, Switzerland, and Ukraine. Experts initially conducted a comprehensive review of the HB-HTA implementation in their respective countries, identifying the barriers and facilitators through descriptive analysis. Subsequently, panel experts ranked these identified barriers and facilitators on a seven-point Likert scale. A median agreement score ≥ 6 and interquartile range (IQR) ≤ 1 was accepted as reaching a consensus. Out of the 12 statements categorized as external and internal barriers and facilitators, the expert panel reached consensus on six statements (two barriers and four facilitators). The external barrier, which achieved consensus, was the lack of the formal recognition of the role of HB-HTA in national or regional legislations. The internal barrier reaching consensus was the limited availability of human resources dedicated to HB-HTA. This qualitative study indicates that HB-HTA still has progress to make before being formally accepted and integrated across most countries, although by building on the facilitating factors we identified there may be an opportunity for the implementation of internationally developed strategies to strengthen HB-HTA practices.

## 1. Introduction

Health technology assessment (HTA) holds a prominent role in facilitating well-informed decision-making process concerning healthcare services coverage, particularly in reimbursing pharmaceuticals [1,2,3] and medical devices [4,5,6]. Typically, national or regional specialized agencies are tasked with conducting HTA within individual healthcare systems [3,7,8,9]. The evolution of HTA since the mid-seventies of the 20th century has led to multiple definitions of HTA reflecting the field’s growth and its interdisciplinary priorities. In 2020, a collaborative effort involving leading HTA networks, societies, and global organizations resulted in a contemporary, internationally accepted definition: “HTA is a multidisciplinary process that uses explicit methods to determine the value of a health technology at different points in its lifecycle. The purpose is to facilitate decision-making in promoting an equitable, efficient, and high-quality health system” [10].

To grasp the essence of this definition, it is vital to acknowledge that health technology encompasses a wide range of interventions tailored for prevention, diagnosis, treatment, health promotion, rehabilitation, and healthcare management [4]. These interventions may take the form of medical devices, pharmaceuticals, vaccines, procedures, programs, or systems. The modern approach to HTA emphasizes a formal, systematic, and transparent methodology, utilizing state-of-the-art techniques to ensure rigor and reliability [11]. Assessing the value of a health technology entails a comprehensive analysis of its overall impact in comparison to existing alternatives; the assessment considers various factors, including clinical effectiveness, safety, costs, economic implications, ethical dimensions, societal and cultural factors, legal considerations, organizational aspects, environmental impacts, and the broader implications for the patients, their families, caregivers, and the population at large [12]. HTA is a dynamic process that spans the different phases of a health technology’s lifecycle, from pre-market approval to disinvestment [10].

There are not only explicit but also implicit factors which are involved in HTA deliberative processes. As far as pharmaceuticals are concerned, in Germany, France, Italy, the United Kingdom, and Spain, these implicit factors have been identified and categorized as the ones related to ethics, psychology, qualification and experience, politics and society, culture, functional role, as well as disease perception [13]. Since there are often ethical considerations associated with implementing HTA in health technology decision making, a concept of procedural justice has been introduced to HTA processes, aiming to arrive at decisions that the public can regard as legitimate and fair [14]. Deliberating on reasons, evidence, and rationale relevant to meeting the health needs of a population should involve a diverse range of stakeholders in a fair and thorough manner [14]. The successful integration and formalization of HTA processes within healthcare systems, coupled with transparent practices accessible to the public, citizens, and taxpayers, play a crucial role in the development and acceptance of HTA. Additionally, international collaboration is pivotal in promoting knowledge sharing and the harmonization of HTA practices [15].

Traditionally, HTA has been predominantly used at a strategic level, involving decision making by entities like states, health plans, and insurance schemes on, for example, the reimbursement of pharmaceuticals. However, HTA has expanded its scope to encompass specific clinical procedures, medical devices, or programs unique to individual healthcare facilities. This expansion has given rise to the concept of hospital-based HTA (HB-HTA) [16], which focuses on assessing the implementation and utilization of health technologies at the operational level within a healthcare system, closer to the patient and healthcare service delivery [17]. HB-HTA can be best described as conducting HTA activities tailored to meet the specific needs of the hospitals, aiming to guide managerial decisions regarding the different types of health technologies, including the processes and methods used to generate HTA reports “in and for hospitals” [18]. HB-HTA is particularly relevant, as hospitals serve as crucial stakeholders and primary entry points for a diverse array of health technologies, which need to be integrated into hospital practices [19].

Gagnon et al. [20] conducted a systematic review and found limited scientific evidence on the impact of HB-HTA on decision making and costs. However, most of the reviewed studies indicated a positive influence of HB-HTA on the introduction or withdrawal of health technologies, receiving positive feedback from managers and clinicians [20]. Subsequently, comprehensive case studies from different countries showed that for HB-HTA to have an impact there is a need to align decision processes across healthcare levels, given the interconnected nature of decisions at these levels [21]. To promote the transparency of HB-HTA, Palozzi et al. [22] proposed a conceptual framework that links clinical, economic, and organizational perspectives while involving stakeholders such as clinicians, healthcare professionals, hospital managers, and patients in the assessment process. Furthermore, a theoretical framework based on multi-criteria decision analysis was developed to aggregate individual expert perspectives when valuing cancer treatments in HB-HTA [23]. Hinrichs-Krapels et al. [24] emphasized the significance of (i) multidisciplinary involvement, particularly from clinical engineers and clinicians, in procurement decision making, and (ii) evidence-based purchasing decisions grounded in HB-HTA. A case study of using rudimentary HB-HTA when considering the introduction of innovative, cost non-neutral technology called for approaches to balance the increased costs with clinical advantages [25], with similar conclusions reached by earlier studies [26,27].

Recent advancements in HB-HTA tools and processes hold the promise of enabling more thorough evaluations of healthcare resource utilization in hospital settings; however, despite its potential benefits, HB-HTA is still not widely implemented in practice. This is primarily due to concerns regarding its complexity and the challenges it poses to healthcare institutions, which often lack the necessary expertise in health economics and struggle to collect and interpret relevant scientific evidence [20,28]. For example, a recent study reporting pilot initiatives in seven hospitals in China concluded that one of the major barriers in the implementation of HB-HTA is a lack of sufficient knowledge of HTA among hospital staff, hindering the effective executions of HTA processes [28]. Similar challenges are notable in Central and Eastern European countries [29].

In 2023, a study was conducted in France among hospitals, focusing specifically on the adoption of innovative medical devices [30]. France showcases a diverse landscape of HB-HTA organizations across various structural levels, particularly within the different categories of hospitals. The study revealed that a majority of the French hospitals acknowledge the pivotal role of HB-HTA processes in guiding decisions concerning the integration of innovative medical devices into their clinical workflows. Nearly all surveyed hospitals have established evaluation mechanisms to assess the feasibility and effectiveness of innovative medical devices.

The study corroborated findings from prior qualitative research in France, indicating that formalized HB-HTA activities were predominantly observed in university hospitals (UHs) [31]. However, the HB-HTA units identified in this survey did not fully align with the criteria outlined in the AdHopHTA project and by Gałazka-Sobotka et al. [32]. Specifically, only one unit had a dedicated full-time HTA expert, and none of the units were currently collaborating with the centralized HTA agencies. This situation was likely attributed to the lack of the formal recognition of HB-HTA in French regulatory frameworks, as well as the absence of official funding for such initiatives.

The adoption of HB-HTA is essential for several reasons. Firstly, it fosters informed decision making crucial for delivering effective and safe healthcare services. By integrating HB-HTA, hospitals gain access to scientific insights and pertinent hospital-specific data, enabling objective- and context-specific assessments, which lead to improvements in patient safety [18,33]. HB-HTA also aids in making more efficient investment choices, helping hospitals save costs by minimizing unnecessary expenditures or avoiding inappropriate investments. The evidence supports the positive impact of HB-HTA on decision-making quality, hospital budgets, procurement, the utilization of new health technologies, and potential costs savings, such as lower purchase prices of pharmaceuticals when considering therapeutic equivalents (biosimilars) or generics in the assessments. Although HB-HTA is well regarded by managers and clinicians [20,34,35,36], its widespread adoption across countries is hindered by the entrenched healthcare system traditions and the historical centralization of HTA [37].

When implementing HB-HTA within a national healthcare system, it is crucial to carefully assess the different implementation models, ensuring alignment with the system’s specific needs and characteristics [32]. In Poland, the efforts to introduce HB-HTA in hospitals drew on international experiences [32]. HTA specialists highlighted that similarly to HTA performed at the national level [7,38], HB-HTA may also benefit from the favorable legislative frameworks; collaboration at international, national, and local levels; appropriate transparent processes; and sustainable funding to promote the rational resource utilization within hospital settings [18].

To gain insights into the facilitators and barriers related to the advancement of HB-HTA, an expert panel consisting of the HTA specialists from seven countries was formed. This panel included three expert groups from Western Europe, three from Central and Eastern Europe, and one from Central Asia. Their collective objective was to reach a consensus that would respect the distinctive features of each country’s healthcare systems. Such a consensus may be particularly beneficial in globally supporting the successful integration of HB-HTA into hospital settings and enhancing the decision-making process, optimizing resource allocation, and improving patient outcomes. Central to the study’s objective was the identification and characterization of the barriers and facilitators influencing the implementation of HB-HTA, considering the evolving nature of methodology and the limited availability of the scientific literature on this subject.

## 2. Materials and Methods

In our study, we focused on seven countries: France, Hungary, Italy, Kazakhstan, Poland, Switzerland, and Ukraine; in each of these countries, we established expert groups consisting of HTA specialists. We aimed to have at least two HTA specialists in each group to reduce potential bias. To be eligible, the experts needed to have expertise in HTA and to be actively involved in HTA as of 6 February 2023. The foundation for this qualitative study was the AdHopHTA manual for HB-HTA [33].

During round 1 of the study, which took place from 6 February 2023 to 27 February 2023, each of the seven expert groups independently analyzed recommendations from the existing HB-HTA guidelines, consensus statements, and literature reviews. They then created descriptive analyses following a defined format covering the following dimensions: (1) the background of HB-HTA, (2) the legal aspects of HB-HTA, (3) the methodology of HB-HTA, and (4) the practical aspects of HB-HTA.

In round 2 from 27 February 2023 to 13 March 2023, three authors (R.D.B., I.L., and R.H.) first reviewed the descriptive analyses of the individual expert groups. Based on these analyses and the AdHopHTA manual for HB-HTA [33], the authors then drafted statements delineating the barriers and facilitators influencing HB-HTA development. These statements were aimed at capturing the collective views and insights of the expert groups. A deliberate decision was made to allocate more time to round 1 than round 2, allowing expert groups within each country an opportunity to thoroughly collect information.

Following the drafting of the statements, the expert members from each country were brought into the international expert panel and asked to indicate their level of agreement with each statement using a seven-point Likert scale, with the scale ranging from one point (strongly disagree) to seven points (strongly agree) [39,40]. The decision was made to employ the widely accepted seven-point scale due to its reliability and validity when compared to a five-point scale [41]. Statements were considered to have reached consensus if they had a median score of at least six and an interquartile range (IQR) up to one [39,40]. An IQR ≤ 1 indicated that more than 50% of the scores clustered within a one-point range on the scale, signifying a high level of consensus on a seven-point Likert scale [42]. This simple method thus allowed for a quantitative assessment of the consensus among the panel experts and ensured that the statements with a high level of agreement and minimal variability among the panelists were identified.

## 3. Results

### 3.1. Participants

A total of fifteen expert panelists participated in both rounds of the study. Among the participants, seven (47%) were female, and four (27%) held a Medical Doctor degree, with two (13%) being practicing physicians; characteristics of expert panelists are summarized in Table 1.

### 3.2. Round 1: Descriptive Analysis of HB-HTA Development along with Identified Barriers and Facilitators

Each expert group in France, Hungary, Italy, Kazakhstan, Poland, Switzerland, and Ukraine prepared a report outlining barriers and facilitators within each of the four dimensions (the background of HB-HTA, the legal aspects of HB-HTA, the methodology of HB-HTA, and the practical aspects of HB-HTA). The outcomes are presented in the following two sections. In Section 3.2.1, we provide an illustrative example of the expert group report focusing on Hungary, while in Section 3.2.2, we summarize the main findings derived from the expert group reports.

#### 3.2.1. An Example of Expert Group Report including Descriptive Analyses of HB-HTA Development along with Identified Barriers and Facilitators for Hungary

##### Background

The current Hungarian HTA system [43] focuses mainly on the centralized assessment of pharmaceuticals and medical devices for reimbursement decisions [44,45]. HTA dossiers are typically submitted by manufacturers, and the assessment included in the dossier is conducted either by the manufacturers themselves or by consultancy firms. The number of HTA experts in the country is sufficient compared to most other Central and Eastern European countries due to previous and ongoing educational programs and the presence of an active scientific society, which is the Hungarian Health Economics Association.

##### Legal Aspects of HB-HTA

The central HTA body in Hungary performs the critical appraisal of the assessments in the HTA dossiers submitted to the National Health Insurance Fund Management (NHIFM). There is currently no governmental organization producing the HTA dossiers. As a result, technologies whose assessments are not initiated by the manufacturers, for various reasons, do not go through the HTA process in Hungary. Most procedures performed in hospitals are therefore not subject to HTA and are reimbursed based on a Diagnosis-Related Group (DRG) [46,47] or fee-for-service arrangements, which have different processes of including new technologies to the reimbursement list.

Innovative medical technologies, which may bring additional benefits to patients, usually also entail higher costs that are not necessarily covered by the DRGs associated with standard care for a particular disease. The introduction of these new technologies in hospitals requires the creation of a new DRG code to cover the additional costs.

##### Methodology of HB-HTA

While there are regularly updated national pharmacoeconomic guidelines in Hungary, there are no additional, specific guidelines for HB-HTA in Hungary. The national guidelines provide detailed and sophisticated advice on the HTA methodology to be used for all assessments. The HTA guidelines cover various aspects, such as comparator selection, the preferred analytical techniques, handling cost and outcome data, modelling, sensitivity analyses, and discounting among several other topics.

##### Practical Aspects of HB-HTA

Hospitals in Hungary have the option to submit a request to the National Health Insurance Fund (NHIF) to create new DRG and fee-for-service codes, but the NHIF may require the proof of value for money. Unfortunately, most hospitals cannot afford to have their own HTA unit, which leads to many innovative technologies remaining unfunded.

Recently, in 2018–2019, two HTA centers were established at two universities in Hungary with large clinical centers (the University of Pécs and Semmelweis University). These centers are faculty departments focused on teaching HTA and health economics modules to undergraduate and postgraduate students. Additionally, these HTA centers develop early HTA models for research units at the universities to support the research and development (R&D) decisions and collaborate with the local clinical centers in the development of HTA submissions for technologies which, by the clinical centers, are considered important for reimbursement. In the latter case, the HTAs are submitted to the NHIF as part of a formal HTA submission. These submissions follow the same procedure as those submitted by the manufacturers and are conducted in accordance with the national pharmacoeconomic guidelines. In the event of a positive reimbursement decision, the NHIF decision will not only entitle the particular clinical center to claim reimbursement for the new procedure, but also any healthcare provider who meets the criteria set by the NHIF.

It is important to note that the role of these HTA centers is not to facilitate decision making at the hospital level (e.g., to purchase a high-cost medical equipment), but rather to work with the university clinical centers to conduct comprehensive HTAs for technologies where no submission from the manufacturers is anticipated. Examples of such technologies include the introduction of modern radiotherapies in cancer treatment [48], invasive EEG to support epilepsy surgery [49], or repetitive transcranial magnetic stimulation for major depressive disorder [50].

The summary findings on the barriers and facilitators for HB-HTA development in Hungary can be found in Table 2.

#### 3.2.2. Summary of Descriptive Analyses of HB-HTA Development

The characteristics of HB-HTA developments, as identified by the expert groups representing France, Hungary, Italy, Kazakhstan, Poland, Switzerland, and Ukraine, are summarized in Table 3. Notably, findings in Table 3 indicate that the current regulatory frameworks in all countries do not formally recognize HB-HTA, leading to a lack of funding dedicated to this activity. As a result, the advancement of HB-HTA has been driven by the hospitals themselves, which showcases the role of the hospitals as initiators and drivers of HB-HTA practices. In fact, some hospitals in France, Italy, and Switzerland were pioneers in the field of HB-HTA [29]. Through their active engagement in international and national initiatives and projects, these hospitals have played a vital role in promoting and fostering the growth of HB-HTA practices.

The descriptions provided in the analyses further indicated that HB-HTA activities in all seven countries have attained a noteworthy level of advancement. The methodological basis and practical recommendations for the implementation of the HB-HTA system have been developed and it is common to all countries so that they can adhere to relevant procedures in conducting HTA. The limited number of HTA experts remains a serious barrier in Italy, Poland, Kazakhstan, and Ukraine, though.

In France, the HB-HTA culture is already well established in most university hospitals and there is a great interest for HB-HTA among other healthcare institutions as well. There was even an initiative launched for creating a community of HB-HTA professionals among French hospitals through the association sf-ets (société francophone pour l’évaluation des technologies de santé à l’hôpital). In Italy, HB-HTA is conducted by a few experienced hospitals that have demonstrated their expertise by participating in numerous international projects (EUnetHTA and AdHopHTA) and networks (International Network of HTA Agencies INAHTA); collaboration between HB-HTA units and academia or research centers is also widespread. In Switzerland, despite early pioneering efforts, HB-HTA implementation at the hospital level remains fragmented, with only two hospitals having an established HTA unit.

In Central and Eastern European countries and Kazakhstan, the interest in HB-HTA has emerged relatively recently, particularly within the last decade. While Hungary and Kazakhstan have acquired some experience in implementing HB-HTA, Poland is currently in the process of actively implementing HB-HTA practices, reflecting the commitment of the Polish National Centre for Research and Development to incorporate this approach into the healthcare system by financing a project »Implementation of HB-HTA in Poland«. Presently, the HB-HTA process is used in several hospitals that took part in this pilot project; however, achieving the critical mass necessary for the nationwide HB-HTA implementation in Poland remains a challenge. Ukraine is still in the preparation phase, indicating early efforts to lay the foundation for future HB-HTA activities with the plan to initiate the HB-HTA process as a pilot project in a single hospital.

### 3.3. Round 2: Expert Panels’ Consensus on Barriers and Facilitators in Developing HB-HTA

Based on descriptive analyses and barriers and facilitators for each of the four dimensions collected in round 1, a total of twelve statements were formulated and organized into external and internal barriers, as well as external and internal facilitators in developing HB-HTA. Among these statements, six received consensus agreement and are highlighted in bold in Table 4.

A common external barrier that reached consensus was the lack of the formal recognition of the role of HB-HTA in national/regional legislations. Additionally, an internal barrier agreed upon was the limited human resources. Another external barrier, though at a moderate consensus level (mean score of six and IQR ≤ 2) was the isolation of hospitals performing HB-HTA. Two additional internal barriers that achieved moderate consensus levels were the lack of support from top hospital management and lack of the involvement of HB-HTA in the definition of hospital strategy. The least consensus was observed for the barrier concerning the potential overlap of HB-HTA with HTA performed at national/regional level.

Among the total of five facilitators, consensus was achieved on four. Two external facilitators were identified as critical: the creation of a network among hospitals performing HB-HTA and the dissemination of HB-HTA methods and activities. In terms of internal facilitators, two key factors emerged: top hospital management support in evidence-based decision making and training initiatives dedicated to HB-HTA. The only facilitator with moderate consensus level was “consensus building in the definition of HB-HTA guidelines and activities”. It is evident that similar facilitators are deemed important in highly developed national healthcare systems (France, Italy, and Switzerland) as in moderately developed national healthcare systems (Hungary, Poland, Ukraine, and Kazakhstan).

## 4. Discussion

### 4.1. The Appreciation and Recognition of HB-HTA’s Role in Healthcare System

Our study has shown that the practical implementation of HB-HTA is still at an early stage in most cases. Currently, HB-HTA is predominantly confined to leading hospitals, such as medical universities and top-tier hospitals, which actively pursue interdisciplinary approaches to make well-informed decisions regarding the integration of innovative technologies. For instance, in Italy, these hospitals frequently participate in scientific research and devote significant attention to incorporating new technologies into clinical and patient pathways.

However, the major external barrier consistently identified across all jurisdictions is the lack of specific regulations that facilitate the formal adoption of HB-HTA and outline corresponding responsibilities. Additionally, the clear operational impact of the healthcare service provision and funding in relation to HB-HTA is not adequately defined. Although the benefits of HB-HTA are acknowledged, particularly by hospital management, healthcare systems have yet to effectively integrate HB-HTA into decision-making frameworks. Coordinated efforts are essential to establish HB-HTA as a valuable tool in advancing value-based healthcare. Hence, it is imperative for healthcare organizers, supported by policy and legal arrangements, to effectively incorporate HB-HTA. It is noteworthy that a one-size-fits-all implementation model may not be feasible due to the substantial differences in healthcare system structures across countries. Engaging multiple stakeholders is vital for the successful integration of HB-HTA into healthcare systems [51,52,53].

### 4.2. HB-HTA and the Practice of HTA in Healthcare Decision Making

In most of the studied countries, HTA has proven effective in shaping national or regional coverage decisions concerning pharmaceuticals, thanks to the presence of specialized HTA bodies that effectively operate and support allocation choices. However, the successful implementation of HB-HTA requires careful coordination to prevent redundancy in activities and responsibilities. Well-defined cooperation models and networking are pivotal to maximize the benefits derived from shared experiences and knowledge. Due to an imbalance between capabilities and expected output, implementing HB-HTA in a single, isolated center is unlikely to be successful. Cultivating collaboration among hospital centers and establishing connections with HTA specialized bodies are crucial steps to enhance the efficient utilization of HB-HTA. Experts consistently emphasize the importance of collaborative support and resource-sharing among HB-HTA centers, particularly in the face of limited human resources, a significant internal barrier.

Building on the concept of joint clinical assessments, efforts should be directed at both the international and regional levels to synchronize HB-HTA. This synchronization is vital to prevent duplication, elevate standards, and assist individual centers or groups of hospitals in addressing specific and critical issues from their perspectives. A common challenge identified in most of the studied countries is the limited capacity and a lack of well-educated and highly skilled professionals in the field of HB-HTA. Given the context of limited resources, promoting the effective and timely utilization of available human resources becomes even more critical. This justifies the necessity for increased networking, collaboration, and educational initiatives, as recommended by experts in our study.

In the backdrop of the dynamic and rapidly evolving healthcare landscape, the multitude of available innovations, and elevated societal expectations, it is essential to establish suitable frameworks for implementing HB-HTA. The failure to do so could lead to missed opportunities and impede the realization of the benefits stemming from HB-HTA practices. Striking a well-balanced trade-off between time, workload, and the depth and complexity of assessments should be prioritized.

### 4.3. Promoting High-Quality Assessment Standards and Best Practice in HB-HTA

The unique character of HB-HTA, shaped by hospital perspectives and political/strategic considerations, underscores the need for a tailored methodological approach while maintaining scientific rigor [18]. In the majority of studied countries, the AdhopHTA manual [33] has been acknowledged as a valuable resource for supporting HB-HTA implementation. It serves as a key guide for developing HB-HTA methodologies whether they are elaborated by hospitals themselves (e.g., Italy and Kazakhstan) or adopted as national guidelines (e.g., France and Poland). It is noteworthy that this manual was developed by the international experts under an EU-funded grant between 2007–2013 (The AdHopHTA Project FP7/2007-13 grant agreement nr 305018) and was officially released in 2015 as a public deliverable of the project. After nearly a decade, there is merit in reconsidering an update to the manual and securing resources for the preparation of a new contemporary version of the manual. It is critical for the revised manual to address, among others, the evolving digital technologies and their rapid integration into healthcare systems, particularly in the aftermath of the COVID-19 pandemic.

### 4.4. Link between National HTA and HB-HTA

The findings of our study underscore a noticeable lack of effective coordination between national HTA and HB-HTA, resulting in insufficient support for high-quality and timely allocation decisions at both the national and hospital levels. The optimal integration model for HTA may vary depending on the healthcare organization, necessitating country-specific arrangements. Both formal and informal collaboration between HTA and HB-HTA is important which is consistent with conclusions from previous research [33]. However, further research is needed to explore this interconnection between national HTA and HB-HTA, aiming to establish a comprehensive ecosystem that facilitates the smooth and efficient implementation of HB-HTA. HB-HTA initiatives can undeniably benefit from the experience and educational support offered by national or regional HTA bodies, as reported for Hungary, Italy, and Ukraine.

### 4.5. Educational and Awareness-Raising Initiatives and Target Audiences

Our analysis of the HB-HTA advancements in the studied countries has revealed predominant educational initiatives primarily focusing on analytical teams at the national, regional, and hospital levels. While this targeted approach is crucial, previous efforts underscore the importance of extending outreach to other key groups, particularly hospital managers and physicians, through awareness-raising educational campaigns [54]. For the effective implementation of HB-HTA, it is necessary to engage these important stakeholders and provide them with a comprehensive understanding of HB-HTA and its potential benefits.

### 4.6. Study Limitations

Our study has several limitations that need to be acknowledged. Primarily, the number of examined countries was limited with varying levels of HTA maturity. Most countries studied (six out of the seven) were from Europe, with Kazakhstan being the only non-European country. Moreover, the participation of the fifteen experts with diverse backgrounds and affiliations in our study may potentially influence outcomes; nonetheless, the inclusion of additional perspectives from a broader range of stakeholders could enrich the study’s breadth. In our study, we engaged the proponents and promoters of HB-HTA from these countries who have a keen interest in fostering development and sharing experiences. By encompassing countries at different stages of HB-HTA development, we created a sample of differentiated experiences from which common conclusions can be derived for the implementation and expansion of HB-HTA.

We recognize that some of the identified facilitators and barriers could be specific to particular technologies. For instance, the legal framework in a specific country might be more relevant to pharmaceuticals rather than medical devices. It is important to further explore these nuances to address the distinct challenges associated with different technologies.

In this paper, we have used an expert panel to assess the status of HB-HTA implementation and identify the potential barriers and facilitators. While this approach may appear to be at odds with more rigorous techniques, such as systematic literature reviews, the existing publications on HB-HTA guidelines are scanty and often the result of the specific academic interest of scholars, resulting in a skewed representation of the countries. Therefore, considering expert evidence is a practical alternative to establish a foundation for our practical document aimed at assisting the implementation of HB-HTA.

### 4.7. Future Developments

The insights from this study can lay the groundwork for future advancements in the field of HB-HTA and may provide impetus for the revitalization of the AdHopHTA initiative [18,33]. This initiative underscores the necessity for more cohesive and extensive investigations into the role and progress of HB-HTA on an international level. The European Commission could take a leading role in propagating the further utilization of HB-HTA among EU member states. Our study is in effect looking at a policy change, and as noted in a change management study [55], effective leadership can sharply reduce resistance to behavioral change including the adoption of innovative methodologies within healthcare.

## 5. Conclusions

This qualitative study, drawing on insights from international experts, achieved consensus by identifying the common barriers and facilitators for HB-HTA implementation. It is important to emphasize that HB-HTA still has not achieved formal acceptance and integration across most countries. In this study, we have pinpointed some of the barriers that still need to be addressed to enhance the adoption of HB-HTA. Further exploration of the connection between national HTA and HB-HTA would be instrumental in precisely defining the role of HB-HTA in the broader healthcare landscape. Following that, coordinated efforts at the European level could help in embedding HB-HTA within national health policies.

## Figures and Tables

**Table 1 healthcare-12-00889-t001:** Expert panelist characteristics.

Characteristics	Number of Experts, *n* (%)
Participation	
Round 1	15 (100)
Round 2	15 (100)
Age, years	
30–39	4 (27)
40–49	5 (33)
50–59	6 (40)
Gender	
Female	7 (47)
Male	8 (53)
Expertise in HTA, years	
<5	2 (13)
5–9	3 (20)
10–14	4 (27)
≥15	6 (40)
Clinical role—Medical Doctor	2 (13)
Expertise (more than one answer can be selected)	
Health economics	14 (93)
Public health	10 (67)
Medical doctor	4 (27)
Pharmacy	5 (33)
Medical or natural sciences	2 (13)
Social sciences	2 (13)
Other	3 (20)
Current work environment (more than one answer can be selected)	
Public healthcare payer	1 (7)
HTA organization	2 (13)
Academia/research institution	12 (80)
Health care provider, including hospitals	4 (27)
Non-governmental organization	2 (13)
Consulting	3 (20)

**Table 2 healthcare-12-00889-t002:** Barriers and facilitators for HB-HTA development in Hungary. HB-HTA—hospital-based health technology assessment; HTA—health technology assessment; DRG—diagnosis-related group.

Hungary	Barriers	Facilitators
Background	HTA is mainly focused on centralized assessments. HTA dossiers are mainly submitted by the manufacturers and carried out either by the manufacturers or consultancy firms. Most of the investments (e.g., the purchase of the high-cost medical equipment) are financed by the government, not by the hospitals, so there is limited need for HTA-based decision support at the hospital level.	The availability of HTA experts is sufficient. The awareness of HTA methods across various stakeholder groups is increasing.
Legal aspects of HB-HTA	Most of the procedures performed in hospitals are not subject to HTA and are reimbursed on a DRG or fee-for-service basis. The introduction of new innovative health technologies in the hospitals may require the creation of a new DRG code to cover the additional costs.	The centralized HTA process has been well embedded in the legal framework of the country for pharmaceuticals and medical devices.
Methodology of HB-HTA	No specific guidance on HB-HTA.	The existence of a regularly updated and detailed national pharmacoeconomic guideline that can be applied to HB-HTA as well.
Practical aspects of HB-HTA	HB-HTA is relatively new in Hungary. The two HTA centers at universities do not aim to facilitate decision making at the hospital level, but rather work with the clinical centers to perform a full HTA for technologies where no submission from the manufacturers is expected.	Experience with HTA in the country is available, in connection with the centralized process since 2004. Two HTA centers have been established at universities with large clinical centers in 2018 and 2019. Large-scale hospital infrastructure projects funded by the EU have in the past required cost–benefit analyses (CBAs). Therefore, expertise is available to hospitals on how to carry out a CBA.

**Table 3 healthcare-12-00889-t003:** Characteristics of hospital-based health technology assessment (HB-HTA) developments in France, Hungary, Italy, Kazakhstan, Poland, Switzerland, and Ukraine. DRG—diagnosis-related group.

Country	Early Initiatives	HB-HTA Processes			Legal Recognition; Collaboration with National HTA Agency	Networking and Advocating
		Methodological Foundation	Assessment Criteria	Participating Hospitals; Evaluation Focus		
France	1982: hospitals’ initiative in Paris region	Initially structured by the hospitals themselves; developed by experts at the request of the Ministry of Health, drawing on the AdHopHTA methodology in 2018	Clinical, economic, organizational, and ethical, including patient perspectives	University and other major hospitals; primarily focusing on pharmaceuticals and medical devices, although other medical technologies included as well	Not formally recognized; national HTA agency not involved in HB-HTA	2022: Society of professionals for HB-HTA networking to promote HB-HTA and foster collaboration with the national HTA agency (still pending)
Hungary	2018: the initiative of two medical universities	The national pharmacoeconomic guideline	Clinical, economic, and organizational, including hospital strategy	A select group of university hospitals and highly specialized hospitals; focusing on medical devices and highly innovative diagnostic and therapeutic technologies, including digital solutions	Not formally recognized; the national HTA agency serving both as a training center and regulator for the approval and reimbursement of the DRGs	Synergy between hospitals performing HB-HTA and research centers
Italy	2006–2012: active engagement in international HTA and HB-HTA projects (EUnetHTA and AdHopHTA)	Drawing on international methodologies, the AdHopHTA handbook, and Core Model	Clinical, economic, and organizational, including hospital strategy	A select group of highly experienced hospitals, recognized as clinical excellence centers; primarily focusing on integrating new technologies into existing medical pathways	Not formally recognized; the national HTA agency as a coordinator of the national and regional HTAs and while also serving as a conduit for disseminating outcomes	Collaboration among hospitals performing HB-HTA and research centers
Kazakhstan	2015: activity undertaken in two prominent hospitals	Developed and published by the leading hospital	Clinical and economic	Two hospitals having dedicated HB-HTA units	Not formally recognized	Early stage of development due to the shortage of trained personnel and financial constrains
Poland	2020: pilot studies conducted as a national grant project »Implementation of HB-HTA in Poland«	2022: »Methodology of HB-HTA« prepared by experts as a draft version, drawing on the AdHopHTA guidelines and insights from pilot implementations in hospitals	Clinical, economic, and organizational	A dozen hospitals primarily from higher reference levels designated as pilot sites; evaluating the different types of medical technologies (diagnostic, therapeutic, and organizational)	Not formally recognized; the national HTA agency serving as the official coordination center for HB-HTA	At an early stage of development, such as training sessions and consultations on the coordination model with the Ministry of Health, National Health Fund, and regional health administrations
Switzerland	2009: national medical board initiative to perform HTA reports in one of the cantons	Drawing on AdHopHTA and aligning with national HTA requirements	Clinical, economic, and organizational	Hospitals participating in international, national, and/or regional HTA projects, with at least two hospitals establishing HTA units	Not formally recognized	National/regional HTA assessments
Ukraine	2021: analyses on legal framework for introducing HB-HTA 2022–2023: study on current decision-making approaches regarding HB-HTA implementation in Ukrainian hospitals	The HB-HTA methodology has been crafted by Ukrainian experts, drawing on the AdHopHTA guidelines and Core model.	Clinical, economic, and organizational	Three hospitals designated as pilot sites for the implementation phase	Not formally recognized. Formal/legal groundwork at an early preparation phase	At an early preparation phase, focusing on training HB-HTA personnel and generating awareness among key stakeholders

**Table 4 healthcare-12-00889-t004:** Barriers and facilitators in developing hospital-based health technology assessment (HB-HTA) with Delphi voting results. MS—median score (1–3 = disagreement, 4 = uncertainty, 5–7 = agreement); IQR—interquartile range.

Statement	MS	IQR
* Barriers—external *		
**No formal recognition of the role of HB-HTA in national/regional legislations**	**6.0**	**1.0**
The potential overlapping of HB-HTA with HTA performed at national/regional level	5.0	3.0
The lack of coordination among the different levels of HTA (macro/meso/micro)	6.0	2.5
The isolation of hospitals performing HB-HTA (Lack of connections among hospitals performing HB-HTA)	6.0	2.0
* Barriers—internal *		
The lack of support from top hospital management	6.0	2.0
The lack of the involvement of HB-HTA in the definition of hospital strategy	6.0	2.0
**Limited human resources**	**7.0**	**1.0**
* Facilitators—external *		
**The creation of a network among hospitals performing HB-HTA**	**7.0**	**1.0**
**The dissemination of HB-HTA methods and activities (i.e., publicly available methodology and free access to HB-HTA reports)**	**6.0**	**1.0**
* Facilitators—Internal *		
**Top hospital management supports evidence-based decision making**	**6.0**	**1.0**
Consensus building in the definition of HB-HTA guidelines and activities	5.0	1.5
**Training initiatives dedicated to HB-HTA**	**6.0**	**0.5**

## Data Availability

The data presented in this study are available upon request from the corresponding author.

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
