# Peer review of "Overcoming Barriers in Hospital-Based Health Technology Assessment (HB-HTA): International Expert Panel Consensus"

_healthcare, 2024, doi:10.3390/healthcare12090889_

Round 1
Reviewer 1 Report
Comments and Suggestions for Authors
This this paper is looking at the barriers and things that facilitate the implementation of with the implementation of hospital-based Health technology assessment in seven countries.
line 98 HB-HTA is still not widely implemented in practice. This is 98 primarily due to concerns about its complexity and the challenges it poses to healthcare 99 facilities, which often lack the necessary expertise in health economics and the capability 100 to collect and interpret relevant scientific evidence.
This paper is in effect looking at a policy change. There has been a lot written about the management of change in the past 30 years by Peter Drukcer and John Kotter and many others, yet no mention is made of this whole approach.
below is a reference to a paper on managing change in the introduction of information systems. This helps to set it in a wider context
“However, effective leadership can sharply reduce the behavioral resistance to change—including to new technologies—to achieve a more rapid and productive introduction of informatics technology. This paper looks at four major areas—why information system failures occur, the core theories supporting change management, the practical applications of change management, and the change management efforts in informatics. “
from
Managing Change: An Overview
,
Journal of the American Medical Informatics Association, Volume 7, Issue 2, March 2000, Pages 116–124, https://doi.org/10.1136/jamia.2000.0070116
Why has HTA for pharmaucuticals been adopted in most ocountieres while HB-HTA has not? What can be learned from how it has been impleement
more refeernces are needed in the introduction. There is only one reference per paragraph, with many statements made without a refence
line 87 MD’s needs expanding. does it mean doctors?
line 101 These concerns are particularly prevalent in Central, Eastern, and Southeastern European countries. Are they particluatly prevalent? The survey was only carried out in the CESEE region so what is known about the level of concern in other countries?
line 136 We aimed to have at least two HTA specialists in each group 137 to minimize potential bias. Doubtful if this minimized bias. It may have reduced it.
table 1 with 15 participants only use whole numbers No decimal places
if there is 0 in that catgory it does not need to be in the table
3.2 Round 1. Descriptive analyses of HB-HTA development along with identified barriers and 168 facilitators in France, Hungary, Italy, Kazakhstan, Poland, Switzerland, and Ukraine 169
Characteristics of HB-HTA developments found by expert groups representing 170 France, Hungary, Italy, Kazakhstan, Poland, Switzerland, and Ukraine are summarized
You don’t need to repeat the list of countries. I suggest remove from the title.
table 2
Cochrane Handbook in Ukraine - please give a reference. Cochrane Hand book I know of is for systematic reivews not HTA
No detail given for Switzerland assessment criteria
line 181 and following. It needs more accurate descriptions. A country as a whole doesn’t usually commit to a policy e.g. commitment of Poland – this need to be more specific. is it the Polish government, the ministry of health, a professional body, some hospital directors or who, has made the commitment? reference if possible
line 219 “While there is recognition of the necessity to adopt HB-HTA.” By whom, reference please
line 270 “has been recognized as an important source of information in the 271 majority of studied countries “
and 276 “manual is widely considered as the gold standard”
these statements need reference otherwise they are opinion
“Our research underscores a noticeable lack of effective coordination between national 285 HTA and HB-HTA, resulting in insufficient support for high-quality and timely allocation decisions at both the national and hospital levels. “ Which research are you referring to here?
line 299 "previous underscores the importance" I think there is a word missing?
line 302 “these important stakeholders” not clear if this refers just hospital managers and physicians, or other groups as well
line 335 “facilitators did not exhibit a correlation with the level of national healthcare system development” I am not clear what this means
Line 336.”It is crucial to highlight that the number of identified facilitators surpassed that of barriers, hinting at potential added value in the formulation of global strategies for HB-HTA.” I am not clear why it is “crucial”. Is this the most important finding?
Is your conclusion not more on the lines that HB-HTA still has a long way to go before it is formally accepted and embedded in most countries. You have identifieds some of the barriers that still need to be overcome and what may help to facilitate its adoption.
line 359 “the assessment is conducted either by manufacturers themselves or by consultancy firms.” Can this be shown to be independent and objective?
line 360 The number of HTA experts in the country is sufficient. How it this assessed
line 364 ‘The central HTA body in Hungary performs an appraisal of the HTA dossiers” This does not agree with line 359. Should line 359 say produced or submitted rather than conducted
Legal aspects of HB-HTA |
Most of the procedures performed in hospitals are not subject to HTA and are reimbursed on a DRG or fee-for-service basis. Introduction of new innovative health technologies in hospitals may require the creation of a new DRG code to cover the additional costs. |
The centralized HTA process has been well embedded in the legal framework of the country. |
The two columns seem contradictory to me. If you have a well embedded centralised HTA process does it not cover the creation of new DRG codes etc for new treatments?
Comments on the Quality of English Languagegenerally good.
Author Response
We wish to thank all reviewers for helpful comments. Please find enclosed our comments.
Reviewer 1
- …below is a reference to a paper on managing change in the introduction of information systems.
As requested by the reviewer, we have added the reference and rewritten the Conclusion section to add potential policy implications (Lines 573-577 in redline version of the manuscript).
- Why has HTA for pharmaceuticals been adopted in most countries while HB-HTA has not? What can be learned from how it has been implemented?
As requested by the reviewer, we have substantially expanded Introduction section in order to address both peculiarities of HB-HTA and challenges the practice is facing. We have added (referring to redline version of the manuscript) Lines 92-102, Lines 124-142, Lines 155-169.
- More references are needed in the introduction. There is only one reference per paragraph, with many statements made without a reference.
As requested by the reviewer, we have added references; Introduction section alone now has in total 38 references.
- line 87 MD’s needs expanding. does it mean doctors?
We wish to thank the reviewer for pointing out this oversight. It means medical devices. We have properly introduced this abbreviation in the first paragraph of the Introduction section (Line 65 in the redline version of the manuscript).
- line 101 These concerns are particularly prevalent in Central, Eastern, and Southeastern European countries. Are they particularly prevalent? The survey was only carried out in the CESEE region so what is known about the level of concern in other countries?
We have rephrased this sentence and added several paragraphs in the Introduction section (Lines 124-142, Lines 155-169 in redline version of the manuscript), which provide broader scope.
- line 136 We aimed to have at least two HTA specialists in each group 137 to minimize potential bias. Doubtful if this minimized bias. It may have reduced it.
We wish to thank the reviewer for this comment, we have made a change as requested (Line 226 in the redline manuscript).
- table 1 with 15 participants only use whole numbers No decimal places. if there is 0 in that catgory it does not need to be in the table.
We have made changes as requested (Line 263 in the redline manuscript).
- 2 Round 1. Descriptive analyses of HB-HTA development alongwith identified barriers and 168 facilitators in France, Hungary, Italy, Kazakhstan, Poland, Switzerland, and Ukraine 169 Characteristics of HB-HTA developments found by expert groups representing 170 France, Hungary, Italy, Kazakhstan, Poland, Switzerland, and Ukraine are summarized You don’t need to repeat the list of countries. I suggest remove from the title.
We have removed countries from the title (Lines 266-257 in the redline manuscript).
- table 2 Cochrane Handbook in Ukraine - please give a reference. Cochrane Handbook I know of is for systematic reivews not HTA
Due to preparatory phase of HB-HTA implementation in Ukraine and changes in the planned approach, which has happened in the interim period, we have adapted Table 3 (Line 399 in the redline version of the manuscript).
- No detail given for Switzerland assessment criteria
We wish to thank the reviewer for pointing out this oversight. Assessment criteria were added in Table 3 (Line 399 in the redline version of the manuscript).
- line 181 and following. It needs more accurate descriptions. A country as a whole doesn’t usually commit to a policy e.g. commitment of Poland – this need to be more specific. is it the Polish government, the ministry of health, a professional body, some hospital directors or who, has made the commitment? reference if possible.
We wish to thank the reviewer for this comment, we have made a change as requested (Line 388-393 in the redline manuscript).
- line 219 “While there is recognition of the necessity to adopt HB-HTA.” By whom, reference please
We wish to thank the reviewer for this comment, we have appropriately toned down the expression (Lines 444-445 in the redline manuscript).
- line 270 “has been recognized as an important source of information in the 271 majority of studied countries “
We wish to thank the reviewer for this comment, we have appropriately rephrased the sentence (Lines 502-505 in the redline manuscript).
- and 276 “manual is widely considered as the gold standard” these statements need reference otherwise they are opinion
We wish to thank the reviewer for this comment, we have removed the sentence (Lines 509-510 in the redline manuscript).
- “Our research underscores a noticeable lack of effective coordination between national 285 HTA and HB-HTA, resulting in insufficient support for high-quality and timely allocation decisions at both the national and hospital levels. “Which research are you referring to here?
We have rephrased the sentence to make it clearer (Lines 502-505 in the redline manuscript).
- line 299 "previous underscores the importance" I think there is a word missing?
We wish to thank the reviewer for pointing out this oversight, we have added the word (Lines 535-536 in the redline manuscript).
- line 302 “these important stakeholders” not clear if this refers just hospital managers and physicians, or other groups as well
Yes, we meant hospital managers and physicians. We have rephrased the sentence accordingly (Lines 537-539 in the redline manuscript).
- line 335 “facilitators did not exhibit a correlation with the level of national healthcare system development” I am not clear what this means
We wish to thank the reviewer for this comment, we have removed this unfortunate sentence from the conclusion and placed its rephrased version under Results section (Lines 431-434 in the redline manuscript).
- Line 336. ”It is crucial to highlight that the number of identified facilitators surpassed that of barriers, hinting at potential added value in the formulation of global strategies for HB-HTA.” I am not clear why it is “crucial”. Is this the most important finding?
Is your conclusion not more on the lines that HB-HTA still has along way to go before it is formally accepted and embedded in most countries. You have identified some of the barriers that still need to be overcome and what may help to facilitate its adoption.
We wish to thank the reviewer for this comment. We have added proposed sentence in the Conclusions section (Lines 584-587 in the redline manuscript) and abstract (Lines 53-54 in the redline manuscript).
- line 359 “the assessment is conducted either by manufacturers themselves or by consultancy firms.” Can this be shown to be independent and objective?
We wish to thank the reviewer for this comment. In Hungary, assessments and critical appraisals are strictly divided. We have modified the text to reduce the potential for confusion (Lines 283-284 in the redline manuscript). Please note that Reviewer #4 requested that Appendix needed to be moved to the main text.
- line 360 The number of HTA experts in the country is sufficient. How it this assessed
We wish to thank the reviewer for this comment. We have added clarification (Lines 285-288 in the redline manuscript).
- line 364 ‘The central HTA body in Hungary performs an appraisal of the HTA dossiers” This does not agree with line 359. Should line 359 say produced or submitted rather than conducted.
In Hungary, assessments and critical appraisals are strictly divided. We have modified the text to reduce the potential for confusion (Lines 290-292 in the redline manuscript).
- The two columns seem contradictory to me. If you have a well embedded centralised HTA process does it not cover the creation of new DRG codes etc for new treatments?
We have added the necessary clarification (Lines 341 in the redline manuscript).
We wish to thank reviewers again for their helpful feedback. We would be glad to send any additional information.
Many thanks and kind regards,
Rok Hren, PhD, MSc IHP (HE)
Corresponding author
Reviewer 2 Report
Comments and Suggestions for Authors
Dear authors, the paper is well written, clear in the explanation of methodology, results and discussion.
Methods of preliminary identification of barriers/facilitators -before Delphi- could be better explained (e.g. keywords, typing of literature review, search strategy).
Author Response
We wish to thank all reviewers for helpful comments. Please find enclosed our comments.
Reviewer 2
- Methods of preliminary identification of barriers/facilitators -before Delphi- could be better explained (e.g. keywords, typing ofliterature review, search strategy).
We wish to thank the reviewer for this comment. We have added clarification (Lines 235-239 in the redline manuscript).
We wish to thank reviewers again for their helpful feedback. We would be glad to send any additional information.
Many thanks and kind regards,
Rok Hren, PhD, MSc IHP (HE)
Corresponding author
Reviewer 3 Report
Comments and Suggestions for Authors
The authors examine the common facilitators and barriers associated with the implementation of hospital-based Health Technology Assessment (HB-HTA) across diverse hospital settings in 7 countries using a qualitative study.
The subject of the article is interesting as well as the adopted methodology. I suggest a minor revision as follows:
Introduction: Please improve this part by adding the following references that may help the reader to better understand the background.
o Monleón C, Späth HM, Crespo C, Dussart C, Toumi M. Systematic literature review on the implicit factors influencing the HTA deliberative process in Europe. J Mark Access Health Policy. 2022 Jun 28;10(1):2094047. doi: 10.1080/20016689.2022.2094047. PMID: 35811835; PMCID: PMC9267410.
o O’Rourke B, Oortwijn W, and Schuller T.. The new definition of health technology assessment: A milestone in international collaboration. In: International Journal of Technology Assessment in Health Care; 2020. doi:10.1017/S0266462320000215
o Daniels N, and van der Wilt GJ. Health technology assessment, deliberative process, and ethically contested issues. Int J Technol Assess Health Care. 2016;32(1–2):10–15. doi:10.1017/S0266462316000155.
Introduction: The authors affirm: “Health technology assessment (HTA) plays a pivotal role in facilitating informed decision-making regarding healthcare services coverage, such as reimbursement for pharmaceuticals and medical devices; national or regional specialized agencies typically undertake HTA within individual healthcare systems”. Please add the references that motivate this part.
Introduction: Please add and clarify the aim of the study and what is the contribution with respect to the existing literature.
Materials and methods: Please motivate why the length of round 1 and 2 are not equal.
Materials and methods: The authors affirm: “Statements were considered to have reached consensus if they had a median score at least six and an interquartile 157 range (IQR) up to one. Please add more references that justify this threshold.
Results: The participants selected for the study are 7. I am not sure that the results are robust. Please discuss better this point in the manuscript.
Discussion: Please move the part of “Limitations” and “Future development “in the conclusions and summarize better this paragraph: there are too many repetitions.
Please discuss the section “Conclusion” in terms of “Policy implications”
A linguistic review is strongly suggested.
Comments on the Quality of English Language
A linguistic review is strongly suggested.
Author Response
We wish to thank all reviewers for helpful comments. Please find enclosed our comments.
Reviewer 3
- Please improve this part by adding the following references that may help the reader to better understand the background. Monleón C, Späth HM, Crespo C, Dussart C,Toumi M. Systematic literature review on the implicit factors influencing the HTA deliberative process in Europe. J Mark Access Health Policy.2022 Jun 28;10(1):2094047. doi:10.1080/20016689.2022.2094047. PMID:35811835; PMCID: PMC9267410.
O’Rourke B, Oortwijn W, and Schuller T.. The newdefinition of health technology assessment: Amilestone in international collaboration. In: International Journal of Technology Assessmentin Health Care; 2020.doi:10.1017/S0266462320000215
Daniels N, and van der Wilt GJ. Health technology assessment, deliberative process, and ethically contested issues. Int J Technol Assess HealthCare. 2016;32(1–2):10–15.doi:10.1017/S0266462316000155.
We have added the requested references.
- The authors affirm: “Health technology assessment (HTA) plays a pivotal role in facilitating informed decision-making regarding healthcare services coverage, such as reimbursement for pharmaceuticals and medical devices; national or regional specialized agencies typically undertake HTA within individual healthcare systems”. Please add the references that motivate this part.
As requested by the reviewer, we have added references; Introduction section alone now has in total 38 references. We have substantially expanded Introduction section and have added (referring to redline version of the manuscript) Lines 92-102, Lines 124-142, and Lines 155-169.
- Please add and clarify the aim of the study and what is the contribution with respect to the existing literature.
As requested by the reviewer, we have added the necessary clarification (Lines 200-213 in redline version of the manuscript).
- Please motivate why the length of round 1 and 2 are not equal.
As requested by the reviewer, we have added the necessary clarification (Lines 240-242 in redline version of the manuscript).
- The authors affirm: “Statements were considered to have reached consensus if they had a median score at least six and an interquartile 157 range (IQR) up to one. Please add more references that justify this threshold.
As requested by the reviewer, we have added references and the necessary clarification (Lines 247-256 in redline version of the manuscript).
- The participants selected for the study are 7. I am not sure that the results are robust. Please discuss better this point in the manuscript.
As requested by the reviewer, we have rewritten Section 4.6 of the Discussion section to clarify this point (Lines 542-553 in redline version of the manuscript).
- Please move the part of “Limitations” and “Future development “in the conclusions and summarize better this paragraph: there are too many repetitions.
As requested by the reviewer, we have rewritten we have rewritten Sections 4.6 and 4.7 of the Discussion section and the Conclusion section.
- Please discuss the section “Conclusion” in terms of “Policy implications”.
As requested by the reviewer, we have rewritten the Conclusion section to add potential policy implications (Lines 573-577 in redline version of the manuscript).
- A linguistic review is strongly suggested.
We have thoroughly scrutinized the entire text and made an effort to improve and refine the style.
We wish to thank reviewers again for their helpful feedback. We would be glad to send any additional information.
Many thanks and kind regards,
Rok Hren, PhD, MSc IHP (HE)
Corresponding author
Reviewer 4 Report
Comments and Suggestions for Authors
The article is dedicated to discovering barriers and facilitators of hospital-based health technology assessment. It imposes to me by its deep Delphi analysis, engaging qualified specialists to conduct the research. The paper is interesting from a practical point of view, especially for understanding fundamental gaps in HB-HTA approach implementation in different countries. This paper encompasses theoretical background analysis and results of an investigation, which, in combination, allow for considerable conclusions that are valuable for the further development of HB-HTA in the healthcare sphere.
But here are some deficiencies that need to be further improved:
1) Keywords to the paper must be specified. For example, decision making – in what sphere; facilitators of what; barriers for what?
2) There is a lack of information about HB-HTA and its peculiarities in the Introduction section of the paper. I strongly recommend to enrich this section with this information. Besides, I think expanding the introduction section with practical examples of using HB-HTA will be appropriate.
3) Line 116 “It is crucial to carefully consider the strengths and weaknesses of different models…” – what models?
4) The paper lacks a Literature review; add that researchers and their works have already conducted scientific studies in the field of HB-HTA.
5) Explain the choice of seven countries for the study. Why this quantity and why these countries?
6) Table 1. Who is the payer (1 (6.7%))?
7) Add text after Table 3. Comment on the obtained results and consensus between different statements.
8) Information about the Hungarian experience in HB-HTA implementation must be moved to the paper's main text. Also, it is advisable to include information about other analyzed countries. In another case, there is a question: why did you analyze this country in detail and not others?
Author Response
We wish to thank all reviewers for helpful comments. Please find enclosed our comments.
Reviewer 4
- Keywords to the paper must be specified. For example, decision making – in what sphere; facilitators of what; barriers for what?
As requested by the reviewer, we have rewritten keywords (Lines 58-59 in the redline manuscript).
- There is a lack of information about HB-HTA and its peculiarities in the Introduction section of the paper. I strongly recommend to enrich this section with this information. Besides, I think expanding the introduction section with practical examples of using HB-HTA will be appropriate.
As requested by the reviewer, we have substantially expanded Introduction section in order to address both peculiarities of HB-HTA and practical examples. We have added (referring to redline version of the manuscript) Lines 92-102, Lines 124-142, Lines 155-169. We have also added references in the Introduction section.
- Line 116 “It is crucial to carefully consider the strengths and weaknesses of different models…” – what models?
We wish to thank the reviewer for this comment. We have rewritten this sentence to make it clearer (Lines 188-193 in the redline manuscript).
- The paper lacks a Literature review; add that researchers and their works have already conducted scientific studies in the field of HB-HTA strategy).
As requested by the reviewer, we have added Lines 124-142 and Lines 155-169 in the redline manuscript.
- Explain the choice of seven countries for the study. Why this quantity and why these countries?
As requested by the reviewer, we have rewritten Section 4.6 of the Discussion section to clarify this point (Lines 542-553 in redline version of the manuscript).
- Table 1. Who is the payer (1 (6.7%))?
As requested by the reviewer, we have clarified in Table that this is a “Public healthcare payer”.
- Add text after Table 3. Comment on the obtained results and consensus between different statements.
As requested by the reviewer, we have rewritten Section 3.3 of the Results section (Lines 416-433 in redline version of the manuscript).
- Information about the Hungarian experience in HB-HTA implementation must be moved to the paper's main text. Also, itis advisable to include information about other analyzed countries. In another case, there is a question: why did you analyze this country in detail and not others?
As requested by the reviewer, we have moved Appendix to the paper’s main text. We have also expanded information about other analyzed countries as recommended by the reviewer (Lines 344-396 in the redline manuscript). We have chosen Hungary as an illustrative example; to include all other countries with the same granularity would in our view cause repetition and the paper would become overly long.
We wish to thank reviewers again for their helpful feedback. We would be glad to send any additional information.
Many thanks and kind regards,
Rok Hren, PhD, MSc IHP (HE)
Corresponding author
Round 2
Reviewer 1 Report
Comments and Suggestions for Authors
This paper has improved considerably. Well done
line 54 although identified facilitators may present an opportunity for the implementation. Not clear what facilitators you are referring to - people or factors
do you mean “although by building on the facilitating factors we identified there may be an opportunity for the implementation “
MD is only used about six times I would suggest writing it in full each time as it only adds 6 words
table 1 still includes figures to 1 decimal place. This is unneceassry and inaccurate. Please round then to whole numbers
30-39 4 (26.7) |
40-49 5 (33.3) |
It is an improvement to have included the text from the appendix A in the main text
line 520 It is crucial to emphasize that HB-HTA still has progress to make before achieving for- 520 mal acceptance and integration across most countries. I suggest “It is important to emphasize that HB-HTA still has not achieved formal acceptance and integration across most countries”
Author Response
We wish to thank the reviewer for helpful comments. Please find enclosed our comments.
Reviewer 1
- This paper has improved considerably. Well done
We wish to thank the reviewer for these comments.
- line 54 although identified facilitators may present an opportunity for the implementation. Not clear what facilitators you are referring to - people or factors do you mean “although by building on the facilitating factors we identified there may be an opportunity for the implementation“
We have made changes as requested by the reviewer (Lines 51-52 in the redline manuscript).
- MD is only used about six times I would suggest writing it in full each time as it only adds 6 words.
As requested by the reviewer, we have written “MD” in full (Lines 60, 73, 102, 144, 148, and 150 in the redline manuscript).
- table 1 still includes figures to 1 decimal place. This is unnecessary and inaccurate. Please round then to whole numbers
As requested by the reviewer, we have made appropriate changes (Lines 229 in the redline manuscript).
- It is an improvement to have included the text from the appendix A in the main text.
We wish to thank the reviewer for this comment.
- line 520 It is crucial to emphasize that HB-HTA still has progress to make before achieving formal acceptance and integration across most countries. I suggest “It is important to emphasize that HB-HTA still has not achieved formal acceptance and integration across most countries”.
We wish to thank the reviewer for this comment, we have made a change as requested (Lines 503-504 in the redline manuscript).
We wish to thank the reviewer again for helpful feedback. We would be glad to send any additional information.
Many thanks and kind regards,
Rok Hren, PhD, MSc IHP (HE)
Corresponding author
Reviewer 4 Report
Comments and Suggestions for Authors
The paper could be accepted in its present form.
Author Response
We wish to thank the reviewer for helpful comments. Please find enclosed our comments.
Reviewer 4
- The paper could be accepted in its present form.
We wish to thank the reviewer for these comments.
Many thanks and kind regards,
Rok Hren, PhD, MSc IHP (HE)
Corresponding author